# What Is a Sarcoma ‘Specialist Center’? Multidisciplinary Research Finds an Answer

**DOI:** 10.3390/cancers16101857

**Published:** 2024-05-13

**Authors:** Roger Wilson, Denise Reinke, Gerard van Oortmerssen, Ornella Gonzato, Gabriele Ott, Chandrajit P. Raut, B. Ashleigh Guadagnolo, Rick L. M. Haas, Jonathan Trent, Robin Jones, Lauren Pretorius, Brandi Felser, Mandy Basson, Kathrin Schuster, Bernd Kasper

**Affiliations:** 1Sarcoma Patient Advocacy Global Network (SPAGN), 61200 Woelfersheim, Germany; 2Division of Hematology/Oncology, Department of Internal Medicine, University of Michigan, Ann Arbor, MI 48109, USA; 3Fondazione Paola Gonzato-Rete Sarcoma ETS, 33100 Udine, Italy; 4Division of Surgical Oncology, Brigham and Women’s Hospital, Dana-Farber Cancer Institute, Harvard Medical School, Boston, MA 02115, USA; 5Division of Radiation Oncology, MD Anderson Cancer Center, Houston, TX 77030, USA; 6Netherlands Cancer Institute, 1066 CX Amsterdam, The Netherlands; 7Department of Hematology Oncology, Sylvester Comprehensive Cancer Center, Miami, FL 33136, USA; 8Sarcoma Unit, Institute of Cancer Research, Royal Marsden Hospital, London SW3 6JJ, UK; 9Campaigning for Cancer, Randburg 2196, South Africa; 10Sarcoma Foundation of America, Washington, DC 20036, USA; 11Sock it to Sarcoma, Perth 6000, Australia; 12Mannheim University Medical Center, University of Heidelberg, 68167 Mannheim, Germany

**Keywords:** sarcoma, specialist center, expert center, clinical expertise, patient advocacy

## Abstract

**Simple Summary:**

A multidisciplinary group researched and defined criteria to describe a sarcoma specialist treatment center or network. The project is led by Sarcoma Patient Advocacy Global Network (SPAGN), involves patient groups from around the world and is supported by clinical specialists and leading researchers. The paper identifies Core Principles and Key Features which define a specialist center/network. They are supported by evidence and experience. A primary aim is that new patients and their families can identify a specialist center which can provide curative treatment. A secondary aim is that countries where specialist treatment is not yet available can identify what they have to achieve to meet that ambition. The Core Principles allow a center or network to attain accreditation as a Sarcoma Intelligent Specialist Network. The Key Features are more aspirational and are expressed in a way that allows local needs, legal considerations and budgetary pressures to be taken into account. This is the first time an expert multidisciplinary group has defined specialism in cancer treatment in a worldwide context.

**Abstract:**

The management of sarcomas in specialist centers delivers significant benefits. In much of the world, specialists are not available, and the development of expertise is identified as a major need. However, the terms ‘specialist’ or ‘expert’ center are rarely defined. Our objective is to offer a definition for patient advocates and a tool for healthcare providers to underpin improving the care of people with sarcoma. SPAGN developed a discussion paper for a workshop at the SPAGN 2023 Conference, attended by 75 delegates. A presentation to the Connective Tissue Oncology Society (CTOS) and further discussion led to this paper. Core Principles were identified that underlie specialist sarcoma care. The primary Principle is the multi-disciplinary team discussing every patient, at first diagnosis and during treatment. Principles for optimal sarcoma management include accurate diagnosis followed by safe, high-quality treatment, with curative intent. These Principles are supplemented by Features describing areas of healthcare, professional involvement, and service provision and identifying further research and development needs. These allow for variations because of national or local policies and budgets. We propose the term ‘Sarcoma Intelligent Specialist Network’ to recognize expertise wherever it is found in the world. This provides a base for further discussion and local refinement.

## 1. Introduction

Over the course of the last 20 years, the treatment of sarcoma in the Western, wealthier, world has improved through the introduction of multidisciplinary management and the development of specialist centers. In some areas, a network approach has evolved with multidisciplinary teams based in different healthcare settings and hospitals, meeting as a multidisciplinary team using information and communications technology.

Much of the world, however, has not been able to adopt these Principles. The population of the world is about 8 billion, and that of Europe and North America combined is only about 17% of that. Data are not available that allow for the total number of treatment centers/networks, or those claiming to be specialist centers, to be calculated. Neither can the quality of the treatment they offer be defined.

It is certain that sarcoma care and treatment standards now delivered in the wealthy world are rarely available beyond it. Patients and patient support groups report this anecdotally. The areas of the world from which patients most frequently reach out for advice about treatment are Eastern Europe, the Middle East, Southeast Asia, Africa, and South and Central America. Doctors providing such advice are mostly from the USA, Europe, the UK, Canada, and Japan. 

The development of the Sarcoma Patient Advocacy Global Network (SPAGN) has brought a patient focus onto this challenge and, consequently, an advocacy consideration of how sarcoma specialism can be encouraged to develop where it does not currently exist and further develop where it does exist has become a priority.

### 1.1. The Need for Specialist Care

Sarcomas are generally poorly understood. They account for slightly more than 1% of all cancer incidence; thus, they are among the rarest, and often least considered, of all cancers. The American Cancer Society’s 2021 GLOBOCAN analysis of worldwide cancer incidence, which has over 46,000 citations, only mentions the specific Kaposi’s sarcoma, ignoring the majority of soft tissue sarcoma and bone sarcoma altogether [1].

Even with effective primary treatment (usually surgery), the reported 5-year survival varies between about 65% in the Western world [2,3] to around 50% worldwide. To complicate the challenge of understanding rarity is the fact that there are over 100 different sarcoma histologies; tumors can be found almost anywhere in or on the human body; and tumors can become quite large and are often painless until they impact organs or patient function.

When sarcoma becomes advanced, treatment options are few. International guidelines rely on toxic chemotherapies such as anthracyclines for many sarcomas, although research with immunotherapies is making small steps forward [4]. Genetic understanding is improving but advances are on a histology-by-histology basis and matching genetic characteristics to available therapies is a highly specialist medical requirement [5]. This latter area of development is exclusively in the wealthier world [6].

Steps are being taken to improve data capture, collation, and analysis. It is recognized that registries offer opportunities to understand and identify areas for improvement. Only in Western Europe are there national registries that can present a comprehensive picture and return analytical data to the front line, whether for policy-making or individual treatment. Even here, a Common Data Model (CDM) [7] is not identified, and cross-border data collation and analysis is unrealized although the ambition is recognized. This work is ongoing [8]. How this can be extended to the developing world is an unknown, although there is growing experience with data integration in areas such as financial services.

### 1.2. Project Rationale

The management of sarcomas in specialist centers is associated with significant benefits for patients [9]. Even though the evidence is limited, it is undisputed that sarcoma patients should be treated by experts in specialist centers; this is regarded as essential in published clinical guidelines [4,10].

Individual patients and the Sarcoma Patient Advocacy Global Network (SPAGN) have used the terms specialist and expert frequently in reviewing sarcoma care and treatment. Some interpretation of the terms can be inferred in these papers [11,12], but a definition has not been addressed.

Although widely used, the terms ‘specialist center’ or ‘expert center’ are rarely defined. They have even become controversial and are being challenged in the light of comments from various parts of the world [13,14,15].

Patient advocacy groups from countries traditionally seen as ‘third world’ also report the absence of specialist care. It is an ambition that needs a means to express the aspiration, introduce the initial standards required, and explain the benefit and value such care can offer.

SPAGN worked with an international group of expert sarcoma doctors to

Identify Features and Principles that define what is needed to manage sarcoma patients optimally.Find a term that could be applied in any area of the world.

We believe that these will help healthcare systems and treatment providers understand the importance of specialist care for people with sarcoma.

## 2. Methods

A discussion paper was distributed to those attending the 2023 SPAGN Annual Conference in Dublin, Ireland, followed by a plenary conference debate, attended by approximately 75 delegates. This resulted in an abstract presentation at the Connective Tissue Oncology Society (CTOS) Conference 2023. Further wider consultation with patient groups and professionals across the world followed. This paper is the result of that work.

### 2.1. Analysis of a Solution

Any terms we choose should recognize that there are national and regional variations beyond the control of sarcoma diagnosis and treatment resources. These differences can be significant and will be impacted by such factors as

Healthcare provision and planning, taking account of population centers or population distribution in rural areas.Budgets, whether these are determined by national provision, hospital constraints or insurance providers.Availability of specific sarcoma expertise (e.g., surgical sub-specialties), even when general oncology expertise is available.Access to specialist resources (e.g., ablative therapy, proton beam).General awareness of sarcoma as a malignancyAwareness of the need to develop and support specialist services.

The meeting attendees agreed that we are looking for high-level Core Principles based on a multidisciplinary approach that can be expressed simply and which can be readily understood. These should be supported by ‘recommended’ or ‘desirable’ factors, which allow nationally specific terms to be used in expressing them.

One non-negotiable Core Principle was accepted by the debate: the treatment and care of all patients diagnosed with a sarcoma should be under the supervision and management of a multidisciplinary team of sarcoma experts with consensus medical decision-making determined at a team meeting or Board.

#### 2.1.1. Accreditation

The challenge of how to certify provision meeting the definition was discussed. There are two elements that are challenging:Inspection. Even using today’s remote systems and with publicly available data, inspection is time-consuming and expensive.Withdrawal of a certification, as such decisions have financial implications and may result in contested decisions.

The Sarcoma Foundation of America has been using a scheme for some years that relies on internal audit and self-certification against a checklist, with centers being reminded that they need to audit afresh when staff change [16]. The Netherlands uses a system that limits the validity of accreditation to 5 years. At that point, an application to renew has to be submitted with the required documentation, including support from patient groups and specialized references.

#### 2.1.2. Networks vs. Centers

There was discussion about ‘networks’ as opposed to ‘centers’. It was recognized that long-standing networks such as those in Scotland or France [17] offer a model that contrasted with the ‘expert single-site’ model used in countries such as Scandinavia, the Netherlands, and the USA. It was also noted that England had started with an ‘expert single-site’ model [18] that has evolved into a mixed system with a ‘surgical center plus local network’ in some regions.

This interdependence of ‘centers’ and ‘networks’ was recognized, and the terms do not compete. Different types and forms of networking were also accepted, ranging from formalized structural relationships to functional and process-led collaborations. Geography may also affect networking decisions. A network approach enables a single clinician with highly specialist skills to work with more than one network if that ensures the best outcomes for patients. It also allows for distributed expertise, which enables consultations closer to home for patients for initial diagnostic work, potentially for some treatment, and for post-treatment follow-up.

#### 2.1.3. Definition of Expertise

There was a need to define the expertise that should be available and provide supporting ideas about how quality can be evaluated and described. Experience and knowledge built through a caseload underpins the learning process that every doctor undergoes. The question of how this can be collectively maximized for patient benefit is a key feature of multidisciplinary patient care.

There is value in a ‘reference network’, with the experience of EURACAN [19] opening up cross-border access to relevant expertise being an example. The point was also made that unintentional effects of regulatory decisions can affect the treatment of rarer diseases. In the USA, medical licensing does not allow the practice of medicine outside the state in which a clinician is licensed, limiting the way that out-of-state specialists can support colleagues or advise patients directly even when they have unique experience to offer. Instead, the patient has to travel to the state where the sarcoma expert is licensed, creating a disparity for those with low economic capabilities, small children, or disability precluding travel. The value of being able to garner advice from a clinician with particular expertise or experience when a specific problem in treatment or care emerges was pointed out, with patients as well as their clinicians wanting to have direct access to such advice.

The role that new technology can play in resolving the challenges of specialist provision is quite clear. Digital pathology and radiology are already positively acting on the diagnostic process. Great steps have also been made as a direct result of the pandemic. A cultural shift that accepts non-face-to-face consultation is apparent in many places, reducing the impact of transport time and costs as a barrier to expert care, although clearly, many clinical interventions still require travel.

Initiatives such as EURACAN also allow for a cross-border consideration of the opportunities opened up by digital technology [7]. Specialist centers and networks generate valuable data and coordinated registries using common data models allow collective analysis, which, in a rare disease, adds value. Supporting the development of expertise in this way is an important part of the clinical future of sarcoma care.

The importance of the MDM/Board Principle in sarcoma care is best evidenced by treatment delays when a multidisciplinary approach is not in place. The evidence indicates that diagnostic delays can be partly attributed to patient ignorance, but they are not patient-driven once concern has been raised [20,21]. The impact of clinical systems and procedures, often involving formal referral, is that they delay patient access to treatment, which the MDM/Board approach reduces [22]. Pressure for faster, accurate diagnosis is a message from patient groups everywhere [23].

A number of further issues for a specialist network or center were raised. They included

Mentorship and training.Communication with the patient is fundamental.Accepting feedback from the patient.A national ‘Gold Standard’ offering a set of ‘base values’.Focus on outcomes and process—well-designed processes reduce treatment delay.Success is multifactorial.

## 3. Results

### 3.1. The Proposed Term for a Specialist Sarcoma Service

Following this discussion, it was clear that the characteristics of the term/phrase we propose adopting mean that it

Can be a co-located specialized team that involves all core specialties, which draws in other specialties when needed and meets as an MDM/Board.May develop a network of providers to reach distant patients so that appropriate care/treatment under guidance can be delivered confidently.Can be a dispersed group of individual expert practitioners who come together regularly to discuss patients (MDM/Board) with ad hoc associated specialists (e.g., from a Reference Network).

The creation of the over-arching term ‘intelligent specialist network’ goes beyond ‘reference’ and ‘expert’. The word ‘network’ indicates that the provision of care does not have to be at a single site. The word ‘intelligent’ reflects the fact that in multi-site provision there is sometimes the need for highly specialized knowledge or skills not usually required and therefore not immediately available within the local network. The type or form of a network will be influenced by local considerations, and it is therefore not defined further. At the same time, the term recognizes that many specialist single-site MDTs will continue to use the term ‘specialist center,’ and the choice of ‘intelligent specialist network’ does not disqualify that description.

We therefore propose that the generic term should be:


**Sarcoma Intelligent Specialist Network**


The context in which it is used will be similar to: ‘people with sarcoma should be treated in a sarcoma intelligent specialist network’.

### 3.2. The Core Principles

A patient’s primary need is for accurate and timely diagnosis followed by safe high-quality treatment, with curative intent, if possible, as close to home as is practicable.

Therefore, the primary and over-arching Core Principle is that:


**The treatment and care of all patients diagnosed with sarcoma should be under the supervision and management of a multidisciplinary team of experts.**


This may be provided through a single meeting or multiple meetings within a team of clinicians acknowledging that they work with one another. All relevant skills that address the needs of the patient(s) being discussed should be present. Presence may be in-person or ‘virtual’ (by phone or video). There should be a regular formal meeting which may be called a Sarcoma Board, Multidisciplinary Team Meeting (MDT Meeting), Team Meeting, or similar. We are using the term MDM/Board in this paper. It is appropriate that where extra-specialist knowledge or skills (e.g., exceptional surgery, treatment of rare anatomic locations, proton beam, nuclear medicine) are required, a relevant member of the team may consult with a provider of such experience/skills, bringing them into the meeting if relevant to do so.

The key characteristic of multidisciplinary management is that formal referral between clinicians is not a requirement; it is a function of team management.

If the meeting is to function as intended, it requires imaging and pathology slides and reports, with the presence of the specialist at the meeting if required. This is indicated in NCCN Guidelines [24] and ESMO Guidance for the treatment of sarcoma [25]. Therefore, there are two further Core Principles:


**Imaging modalities must be available with MRI as appropriate.**



**An experienced sarcoma specialist pathologist should either be the primary reviewer of the biopsy/tissue sample or the provider of a second/confirmatory opinion.**


### 3.3. Who Should Be an MDM/Team Member?

The multidisciplinary team approach relies on all patients being discussed and consensus treatment decisions being reached through the consideration of all relevant clinical expertise. This is especially important at the initial stage following diagnosis, when curative intent is the primary aim and adjuvant or neo-adjuvant therapies are considered.

Multidisciplinary patient management is demonstrated through formal meetings of the whole team on a regular basis with individual patient decisions being recorded and the ‘next step’ clinical responsibility decided. This MDM/Board meeting will have a locally recognized name and will observe clinical guidelines that have national or international recognition. There will be instances where smaller, less formal meetings between clinicians make certain treatment decisions, but these should be brought to the full formal meeting for confirmation.

The membership of the formal multidisciplinary team should be disclosed and available to patients. In each case below, the * (starred) member should be an experienced sarcoma specialist, and this is a mandatory requirement. They are

*Surgeon(s) with specialist experience in treating sarcoma.Surgeon(s) with special interest, e.g., retroperitoneal, head and neck, endoprosthetics, thoracic.*Histopathologist—access to molecular pathology.*Radiologist—imaging/diagnostic.*Oncologist with radiotherapy expertise.*Oncologist (medical) with drugs/medicines expertise.Oncologist (pediatric) with experience treating children/adolescents and young adults (AYA).Radiologist—interventional.Palliative care specialist physician.Sarcoma specialist nurse (see Note 1 below).Physiotherapy/rehabilitation practitioners.Fertility and sexual health especially with AYA.Psychologist.

Other clinical expertise should be brought in as required if they are not among the special interests or experience of those above. These areas of expertise may include (not an exclusive list)

Ablative therapies.Medical physics/nuclear medicine (including PET).Surgical sub-specialties and expertise (including robotics).Novel therapies (including immunotherapy).Genetics.

Recommendation


**The Sarcoma Intelligent Specialist Network must maintain contact with providers of clinical services that are used relatively infrequently, regularly review such contacts, and have clear methods for bringing in such expertise.**


The leader of an MDM/Board will be a senior clinician. Succession planning is not always a feature of clinical management, and quite often, succession is left to chance. For stability and to ensure that a Network continues to meet all the requisite Principles and Features, we have a further Recommendation.

Recommendation


**The Specialist Network can identify suitable locum expertise, and when a specialist member of the clinical team moves on, quickly fill the role and ensure that the level of expertise offered to patients does not diminish.**


**Note 1:** The value of the sarcoma nurse specialist has been shown, and patients regard sarcoma nurse specialists as essential. One of the roles is to support patients through the diagnostic and primary treatment process. Where there are national practice issues that make such nursing provision problematic, this role should be highlighted as essential, and its provision should be an aspiration wherever possible.

Recommendation


**Sarcoma specialist nurse(s) are in post and seen as key members of the team.**


### 3.4. Additional Features of a Sarcoma Intelligent Specialist Network

It became clear during the discussion that patient needs can be defined in four features of a Sarcoma Intelligent Specialist Network:Expertise.Knowledge.Resources.Access.

The label of Sarcoma Intelligent Specialist Network should feel aspirational. The following are Recommendations that may be adapted to address national, regional, or local considerations, especially when budgets are considered.

1.Expertise.

It is accepted that the caseload a Network handles will underpin the way its expertise develops and grows [26,27]. Therefore, a high case volume should be a mandatory requirement for Sarcoma Intelligent Specialist Networks.

It also indicates that reducing diagnostic delays should be a target activity for the Network [28]. Educating professional colleagues in primary and secondary care can speed up referrals, leading to earlier access to curative treatment. Opportunity should be taken to gain general media coverage so that the wider public’s understanding of sarcoma also improves, with the aim of reducing patient-initiated diagnostic delays.

The incidence of sarcoma is poorly understood. Overall incidence rates of 70–80 per million of the population come from Europe, while the USA records about 40 per million. Patient stories tell us that there will be late diagnoses with advanced disease, some patients will come through referral after treatment which reveals sarcoma (e.g., uLMS), and some will be GIST, going straight to a medical therapy from a GI specialist. Thus, it is not easy to estimate the sarcoma caseload for a Network and the scale of the population that would support it.

In the USA, the Sarcoma Foundation uses a caseload of 50 patients per annum as the minimum to establish expertise. The same applies in Germany, with the threshold defined by the German Cancer Society [29]. To reach this would probably require a population of 2–2.5 million. The economic calculations used for England indicated a base of 5 million people to justify a specialist center, although this is distorted because some of its centers have a national as well as a regional/local role. It set a minimum standard of 100 cases per year, and one center that had been initially authorized was closed after five years for low numbers. In Italy, the national recommendation is for 100 soft tissue and 50 bone cases each year.

There have been data presented in TARPSWG (Transatlantic Australasian Retroperitoneal Sarcoma Working Group) meetings on surgery for retroperitoneal sarcoma that support a caseload of 24 per annum to define expertise in treating this specific sarcoma patient group. Such work supports our approach to using caseload as an indicator of expertise. Local decisions on caseload for pediatric units would need to be made.

It is unlikely that single institutions in small countries with a small population will be able to treat this number of cases, so thresholds should be realistic but not reduced to accommodate more Centers with less expertise. Thus, a nation with a population <5 m could be appropriately served by one Network structured to meet its geographical realities.

Recommendation


**A Sarcoma Intelligent Specialist Network should have a caseload of 100+ sarcoma patients each year, with 50 newly diagnosed patients, to assume appropriate sarcoma expertise.**


2.Knowledge.

Knowledge is best gained through initial training and continuing education. The contact that a Network’s clinicians have with other sarcoma specialists through formal opportunities such as exchanges, fellowships, training, oncology conferences, and other events is important. The active involvement of surgeons and oncologists undergoing training or immediately following initial qualification should be part of the Network’s routine function. Those wishing to follow a pathway to sarcoma specialty should be encouraged.

Recommendation


**The Network should have a clearly stated priority that it is a ‘learning organization’ committed to ensuring that all its staff have the initial training they need and that it supports all kinds of training opportunities so that they gain the knowledge needed to provide optimal sarcoma management.**



**The Network should be a member of regional or national sarcoma specialist interest groups, attend meetings, and share training opportunities.**



**Clinicians should be members of international sarcoma organizations such as CTOS and specialty working groups such as TARPSWG, and associated with research groups such as EORTC (European Organisation for Research and Treatment of Cancer) and oncology organizations such as ESMO (European Society for Medical Oncology), and they should attend meetings.**


3.Resources.

We do not offer detailed provision of technical equipment, so we discuss the overall picture of emerging technology. Radiographic imaging in particular is rapidly evolving, and so too is access to specialized techniques such as isolated limb perfusion, ablative procedures, or proton beam therapy. This technology will ensure a high level of quality care for sarcoma patients.


**A Sarcoma Intelligent Specialist Network should have access to new technology and imaging techniques as well as to specialized treatment options.**


Molecular Pathology and Genomic services: This is an area that is fast developing. The case for whole-genome sequencing at first diagnosis is being actively argued. The potential for new adjuvant therapies and possibly maintenance therapies is becoming clearer. Meanwhile, there is an established case for understanding the genomic mutations present in advanced disease as the science of treating by histology + mutation or by mutation alone, rather than by histology alone, is rapidly advancing. There is growing evidence of the value of immunology treatments, generally in combination with standard therapies, in specific subtypes of sarcoma. There is already evidence in select subtypes (notably GIST) where sequencing should be performed prior to the initiation of systemic therapy. It is likely that the number of subtypes and relevant new therapies will grow. Collaboration between the oncology and pathology specialists in a Network is essential to ensure that opportunities are not missed for individual patients.

There is also growing knowledge of germ-line mutations such as the *P53*-related Li Fraumeni syndrome which if diagnosed in a family has significant implications.

Recommendation


**A Sarcoma Intelligent Specialist Network should have access to a genomic analysis service and supporting counseling and psychological expertise.**


Advanced Information Technology: A Network will be using the latest communications technology to facilitate meetings, one-to-one conversations, and share files (images, etc.). A stable infrastructure is necessary and usually will be provided as a core hospital service. How this might extend to those outside the institution needs to be assured.

Gathering quality data is an important task for clinical record-keeping, analysis, and decision-making. It is widely understood that improvement on the basis of analyzing good data is a key feature of developing a beneficial and valuable patient service in every aspect of clinical care. Systems should be planned and coordinated, and staff should be trained and must understand the importance of data quality.

Registries are a major consideration, although they have high set-up costs and many technical issues to resolve as well as ongoing costs. Collaboration is a key feature for registry development in rare diseases, allowing larger multi-site and pan-national datasets to be available. Such registries need to be based on a Common Data Model so that easy and rapid sharing of data is possible. Such models do not exist for sarcoma at present although work is underway in the EURACAN Blueberry project [8]. International collaboration will open up the sharing of data and allow access to academic analysis which can bring additional expertise. Patient data confidentiality is now enshrined in the legal statutes of most countries, and some states give special consideration to research using anonymized data from patient cohorts.

At this point, we have no specific Recommendation to make with regard to AI other than to use caution. AI has demonstrated that it has the potential, particularly in diagnosis, to give radiologists valuable advice where there is uncertainty. National and international projects are taking these issues forward, so our recommendation is simple.

Recommendation


**The Network ensures that all relevant clinical data are recorded and that national standards for data quality, data sharing, cancer registration, and patient confidentiality are observed.**


4.Access.

Promoting good quality and safe care of patients across the Network should be a primary aim. While the Network will aim to assure the quality of its processes it is equally important to assure the quality of each patient’s journey through the care pathway. The EU Joint Action on Rare Cancers introduces the concept of the ‘network patient’ who is treated according to the Network’s clinical practice guidelines, exploits the clinical expertise available in the Network, and is registered in the Network database [30].

Similarly, the concept of ‘shared decision-making’ is increasingly recognized as important for the effective treatment of patients, especially when complex decisions that can impact quality of life are required [31].

Recommendation


**The Sarcoma Intelligent Specialist Network assures a high quality and safe pathway for each individual Network patient observing the Principles of ‘shared decision-making’.**


People with sarcoma ask for access to high-quality and current treatment options but also to research and clinical trials. Participation in scientific (laboratory) research usually requires access to an academic center but can be supported in many ways by enrolling patients in tissue sampling and banking, non-interventional real-world research cohorts, or data sharing, for example.

Clinical trials are carefully regulated and require supporting resources such as research nurses, tissue collection, operational managers, methodologists, and medical statisticians. Clinical research is also an opportunity for the study of care using patient-reported outcomes (PROs). There is evidence that hospitals where clinical research is conducted have an overall better outcome performance than those not conducting research [32].

Where appropriate, it is important for patients to be offered the opportunity to enter a clinical trial. For some, it will be a route to extending their life. Most sarcoma clinical trials are multi-site, whether sponsored by industry or academic institutions.

Employing real-world clinical and registry data as a source for research is growing, emphasizing the importance of gathering high-quality data. The use of real-world data as a comparator for research in place of a placebo comparator is also evolving; placebos are widely disliked by patients.

Recommendation


**A Sarcoma Intelligent Specialist Network should offer patients the opportunity to participate in research whether non-clinical or clinical.**



**The Network should have access to clinical trials available in its national context and should seek to offer such studies to appropriate patients or refer patients to a center offering relevant studies.**


Patient Experience

Patient-reported outcomes are becoming generally recognized as adding value to research. The increasing availability of sarcoma-relevant PRO tools is allowing evaluation of patient experience and satisfaction along the entire patient pathway. While we have no information that indicates these tools are currently being widely used in this way, we believe that patient experience and assessment can exert a direct impact on a Network and help raise the quality of the outcomes it achieves.

Recommendation


**A Sarcoma Intelligent Specialist Network collects and analyses data on patient experience and uses them within its self-audit and improvement planning.**


Patient Support Group

There is real value to patients in a patient support group, or groups, associated with the Network. The quality of the patient experience is improved by exchanging experiences with those who have faced something similar. This may be through social media but is enhanced by face-to-face contact.

Recommendation


**The Network should facilitate patient support groups in the Network while encouraging patient leadership and management of such groups.**


Groups may come together at a national level to allow a single patient voice to be expressed in the national healthcare system and to build relationships with groups in other countries through membership in the SPAGN. Ideally, each site or network should have a sarcoma-specific patient support group that can provide support to patients in the context of local needs, provide advice to both patients and the MDM/Board when required, and help its national group in developing the patient voice.

One valuable activity for a national group would be to work with the specialists in their national Network(s) to gain media coverage of sarcoma-related stories—whether reporting patient experience, locally conducted research, or individual stories of interest. This way, the general public’s understanding of sarcoma can be raised.

Patient advocacy groups may also develop research projects, sometimes working with clinical professional partners, sometimes looking at non-clinical issues on their own. Patient-led research is a development that should be encouraged. There is also a growth of evidence-based advocacy, which can bring patient-led data to the fore in decision processes and policy-making. A strong example is seen in the work of SPAGN and the Netherlands Cancer Institute on the PPRN Priority Setting Partnership [33].

## 4. Further Discussion Points

Further points were made in the discussion, which, although not specifically addressed within the main text, should be noted. They create part of the underlying ‘tone’ and ambition of this work.

These points:

The patient wants a timely accurate diagnosis and to know/understand treatment options.

Patients should be told if there is a treatment center closer to home.

Look at outcomes and not processes but recognize that good processes deliver good outcomes.

Develop intelligence through research, gathering and using data that are both patient- and activity-oriented.

Be clear about the ‘added value’ specialists bring to patient care.

Recognize realities and be practical and creative.

## 5. Conclusions

We propose that the term Sarcoma Intelligent Specialist Network supersedes the use of ‘specialist center’ or ‘expert center’ with regard to sarcoma treatment. Where there is a single-site provision of multidisciplinary team management, the term ‘Center’ may be applied as a descriptor instead of ‘Network,’ but any supporting description should make clear that the Network standard is observed.

This work will continue and evolve under the management of SPAGN, consulting with patient group members and clinical leaders as appropriate. While we hope that each team treating sarcoma will wish to follow and implement the Sarcoma Intelligent Specialist Network Principles and Recommendations, we also hope that healthcare systems and funders will take note and consider appropriate supportive actions.

The importance and value of data gathering; good, well-designed data systems (using a Common Data Model); and sharing data for analysis cannot be understated. This is one area where the early implementation of good practices in a newly developing Sarcoma Intelligent Specialist Network could have the effect of rapidly accelerating improvement and change, creating care and treatment opportunities that benefit patients.

Self-accreditation is the proposed route forward. The Network Standard is composed of the Core Principles and Features discussed above. They are summarized in the attached Checklist in Appendix A, which could be used for a self-inspection process. It is suggested that self-inspection is performed by the clinical team working with representatives of their patients. The patients will question and suggest, make the clinical team think twice, and be able to take items from the Checklist into their own agenda. Each of the items in the Checklist may be ‘scored’ at the discretion of the team. The Core Principles should be regarded as non-negotiable—if they are not achieved, the Network description cannot be applied. This is the only absolute requirement for the use of the term Sarcoma Intelligent Specialist Network.

## Data Availability

Discussion papers referred to in the document are available on request.

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
