# Peer review of "What Is a Sarcoma ‘Specialist Center’? Multidisciplinary Research Finds an Answer"

_cancers, 2024, doi:10.3390/cancers16101857_

Round 1

Reviewer 1 Report

Comments and Suggestions for Authors

Dear authors,

I appreciated the paper proposed.

In this deepen and well-structured analysis on specialized sarcoma centers, there is some missing points in my opinion that should be discussed to improve future activities in sarcoma specialized centers.

These are my suggestions for an implementation of discussion in the paper:

A specialized sarcoma center, should not only educate sarcoma specialists, but should work to educate the physicians of the territory (generalists, internal, radiologists) to recognize this conditions as fast as possible. This is a key-element, that is missing in the current sarcoma specialized centers missions (at least to the best of my knowledge). The main goal of this continuous education is to reduce the diagnostic interval and to speed-up all the therapies - I suggest You to discuss this (suggested references PMID: 36524611 - PMID: 36524611).

Moreover, I think that you should briefly discuss how this education of non-specialists physicians could be performed. I think that this should be performed in two main ways: (i) organizing simple informative congresses for non-specialized physicians - (ii) publishing educative and/or clinically useful paper in general journals and/or paper focused on early disease detection. In regard to this latter point it should be underlined and discussed the key-role of radiologists (even non-sarcoma specific ones). Radiologists are the first "screening" person that (with US and or MRI - PMID: 36868902) can recognize a possible malignant (sarcoma) condition, even in very rare diseases (suggested ref. and example of research of this kind PMID: 37987424).

Author Response

Thank you for your comments. They are adding value to the paper.

Educating the physicians who handle first line care, family doctors and others in general practice, is a primary concern at present (as evidenced in refs xx). Our paper should reflect that and the mission to reduce diagnostic delays and speed up access to curative therapy in particular. We have added a paragraph and a supporting reference.

The education of non-specialists is challenging. Today doctors have to handle a fast-changing range of scientific and medical understanding and it is too easy to dismiss sarcoma as “only 1% of all cancers”. Access to general journals has proved difficult for those writing about sarcoma – the “only 1% of all cancers” attitude strongly persists. We will not stop trying and in the meantime the public media coverage of sarcoma is increasing as patient-led support activity increases. We hope this will held engender change. We have added a comment reinforcing the importance of reaching media of all kinds with the educational message as a role for both the Network and for patient advocacy groups.

The key role of radiologists is accepted and understood. Balancing all the key roles, especially radiology and pathology along with a clinical understanding of the patient, is a function of the MDT while in our paper we are trying to find a balance for a more strategic overview, rather than guidance for practice. I accept the case put forward in the papers you refer to, but I must add that there are similar views from other professional standpoints. What we are trying to offer is a neutral background which allows each MDT to develop its skills and knowledge.

Reviewer 2 Report

Comments and Suggestions for Authors

Manuscript entitled "WHAT IS A SARCOMA ‘SPECIALIST CENTER’? Multidisciplinary research finds an answer."

As a pathologist devoted in the molecular diagnosis of sarcoma, I appreciate the efforts of the authors. This work is meaningful and sound. It can be accepted pending revision:

1. The authors should disclose the current number and quality of "SARCOMA ‘SPECIALIST CENTER" worldwide to justify the unmet need.

2. I would suggest the role of molecular pathology should also be included into the center.

Comments on the Quality of English Language

Acceptable

Author Response

The number and quality of Sarcoma Specialist Centers worldwide is one of the challenges we are hoping this paper will help reveal. Even in the well-resourced world of western Europe, USA, Canada and Australia/New Zealand, it is hard to define a precise number of Centers and ‘quality’ is a variable. We have emphasised this in the Introduction.

For information:

An example is Scotland which has five centers but not all are equal – all cover diagnosis/radiotherapy/chemotherapy, three do surgery but only one does retroperitoneal surgery, there are just two medical oncologists. Pediatric cases are managed in specialist paediatric units/hospitals but surgery will be with a sarcoma specialist from one of those three centers. They work as a network.

 In the USA the number of centers is fluid as specialists move/retire etc. There are US centers which are self-defined as specialist and patient groups can be wary of recommending them.

In India there are specialists but they rely on the support of generalists for some treatments, and there are not enough of the specialists to allow easy access to their treatment across the whole country. In South Africa there are no specialist doctors, although there are some who have “an interest”.

Our analysis indicates that just 17% of the world population has reasonable access to specialist treatment and care. That is what we must endeavour to change.

I accept your comment about molecular pathology. It has become a fast-moving scene and although the case for reporting molecular characteristics of sarcomas at first diagnosis has not yet been generally accepted it is strengthening. Its value in more advanced disease where immunotherapy becomes a potential treatment and targeted therapies are developing is not denied. We have strengthened those paragraphs in the paper.

Reviewer 3 Report

Comments and Suggestions for Authors

The Authors aimed to offer a definition for patient advocates and a tool for healthcare providers to underpin improving the care of people  with sarcoma.

The topic is interesting but the paper is disorganized and very difficult to follow.

It is not clear if this was an attempt for a consensus meeting or something similar.

Data regarding the Professionista involved are necessary (speciality, nationality/workplace).

In my opinion, an epidemiological paper which includes the composition of MDTs around the World, as well as the definition of "expertise" among different Countries would be of paramount importance. 

Conclusions are not supported by results.

Comments on the Quality of English Language

Many grammar and syntax errors.

Author Response

I am glad this reviewer found the paper interesting but saddened that he felt it was disorganised and difficult to follow. Some of this may be due to re-formatting into the review version.  I acknowledge that in merging various sections in the developing paper in order to bring the final paper to a reasonable scale some of the character of having individual sections has been lost.

This was not an attempt at reporting a consensus meeting although there was strong consensus on many elements of the case being made during the open meeting at the SPAGN conference in 2023. Different sections of the paper then relied on different sources and constant review was necessary to keep them aligned on the Core Principles, which was the key consensus element.

I do not understand the point about ‘data regarding the ‘Professionista involved’. There were 75+ patient advocates in the meeting referred to above. These included active sarcoma patients, advocates with 20+ years of experience working with professionals and patients, and some of these people work in professions such as law, teaching, journalism, accountancy etc. They are lay people. The professional medics involved as co-authors are all prominent sarcoma specialists and there were many more in almost every related medical discipline who read, commented and contributed in smaller ways. It is impossible to count the number of nationalities involved. As far as we are aware this is the first paper ever developed in any cancer which has global cross-cutting support for a clear objective.

The comments seems to have missed the objective of the paper. An epidemiological paper would go deeper into the problems, which are different in every healthcare system, but would not offer a solution.  We have addressed the issue of numbers in the Introduction. Only some 17% of the world population is covered by a sarcoma MDT. There is a lot of work to do getting the Core Principles adopted and contributing to global change.

As patients we were determined that this paper should be written in ‘plain language’. This required a more colloquial approach, which does not always conform with grammatical and syntactical perfection. I do not deny such errors exist but I feel that taking the ‘plain language’ approach makes the paper more accessible to those non-scientific readers as well as doctors. These include patient advocates, policy-makers, funders and healthcare managers.

Round 2

Reviewer 2 Report

Comments and Suggestions for Authors

The revision is well performed.

Comments on the Quality of English Language

Acceptable.

Reviewer 3 Report

Comments and Suggestions for Authors

Despite the Authors'efforts, they were not able to address appropriately to most of my previous concerns.

The paper lacks any scientificity and it is still very disorganized and difficult to follow. 

Comments on the Quality of English Language

Many grammar and syntax errors.